# EDGE PARTITION MODULATED GRAPH CONVOLUTIONAL NETWORKS

## ABSTRACT

Graph convolutional networks (GCNs), which propagate the node features through the edges and learn how to transform the aggregated features under label supervision, have achieved great success in supervised feature extraction for both graph-level and node-level classification tasks. However, GCNs typically treat the graph adjacency matrix as given and ignore how the edges could be formed by different latent inter-node relations. In this paper, we introduce a relational graph generative process to model how the observed edges are generated by aggregating the node interactions over multiple overlapping node communities, each of which represents a particular type of relation that contributes to the edges via a logical OR mechanism. Based on this relational generative model, we partition each edge into the summation of multiple relation-specific weighted edges, and use the weighted edges in each community to define a relation-specific GCN. We introduce a variational inference framework to jointly learn how to partition the edges into different communities and combine relation-specific GCNs for the end classification tasks. Extensive evaluations on real-world datasets have demonstrated how the proposed method helps enhance the discriminative representation power and its efficacy in learning both node and graph-level representations.

## 1 INTRODUCTION

The graph data structure is widely adopted to describe interconnected objects, such as users in a social network and atoms held up by chemical bonds to form molecules. The edges in real-world graphs usually indicate strong and logically reasonable relations between nodes. Graph convolutional networks (GCNs) (Defferrard et al., 2016; Kipf & Welling, 2017; Veličković et al., 2018; Klicpera et al., 2019a;b) capture this property of graph data by introducing neighborhood aggregation into regular perceptron layers. While this approach typically enhances the feature dependencies among strongly-related adjacent nodes, the absence of relational inference makes the aggregation mechanism indifferent to the multiple types of latent inter-node relations, ignoring which limits the ultimate potential of the model performance.

Extensive efforts have been made to seek solution for performance enhancement in dealing with graphs with heterogeneous relations (Schlichtkrull et al., 2018; Vashishth et al., 2020; Nickel et al., 2015), where the common ground these methods converges to is to learn multiple graphs, each one specified to one type of relation. The separation of relations-specific graphs from one graph is easy when these relations are explicitly annotated by edge labels, but is non-trivial when the relations are non-observable. To address this issue, some GCNs introduce techniques such as multi-head (Veličković et al., 2018; Gao & Ji, 2019) or multi-hop (Abu-El-Haija et al., 2019) aggregation that increase the model capacity for modeling multi-relations, however, the aggregation weights in these models are either short of interpretability or too dependent on human knowledge to adapt to the underlying relations. Another line of research aims to disentangle relation-specific node representations by task-driven graph factorization (Ma et al., 2019; Yang et al., 2020), where the relational disentanglement is performed relying on label supervision, hence their performances would be negatively impacted by the scarcity of observed labels.

In contrast to the aforementioned methods, we introduce *edge partition modulated graph convolutional networks* (EPM-GCNs), a generative framework that models the generation of edges and labels from latent relations. To begin with, we represent the latent relations with co-membership of $K$ overlapping latent communities. For instance, in a social network where people are linked

by online friendship, one person may be simultaneously affiliated with multiple social groups (e.g., graduating from the same school A, or working for the same company B, etc.). Then some similarity-based relations (e.g., schoolmates of A, colleagues in B) would naturally exist among people that are affiliated with the same social group. Under this modeling assumption, the strength of each relation between a pair of nodes is measured by the multiplication of the following latent quantities: (i) the average interaction rate among co-members in the corresponding community and (ii) how strong the two ending nodes are affiliated with this community. The edges are further modeled by the logical OR of binary latent edges independently generated through the Bernoulli-Poisson link function given the strengths of each relation. This part of the generative model explains the shrinkage of heterogeneous relations into binary latent edges.

We next build up the generation process of labels. Since one node simultaneously engages in multiple relations, we generate its representation as a composition of relation-specific sub-representations. These sub-representations are learned by three steps: the first step partitions the edges by normalizing the strengths of $K$ community-based relations, and defines $K$ independent GCNs with each set of weighted edges; the second step obtains the desirable sub-representations by running the $K$ GCNs in parallel. The labels of nodes or graphs could be predicted with the node representations generated in this way. This generative model allows the community-based relations to be inferred from both labels and edges, which not only systematically solves the relation non-observable issue, but also incorporates edges as an alternative source of information provided for the posterior inference of latent relations. Since the observed edges dominate labels in quantity, our method would not heavily depend on label supervision, hence may suffer less from the scarcity of labels.

We summarize the major contributions of EPM-GCNs as follows:

- We develop a novel generative model that generates labels from relation-specific sub-representations, which are at first learned through independent neighborhood aggregations with weighted edges that are partitioned by the strengths of latent relations;

- We integrate the label generation process with the generation of edges, which enables the observed edges to compensate for the scarcity of observed labels by providing extra information for latent communities/relations;

- The proposed EPM-GCNs are end-to-end trainable via variational inference;

- We analyze the working mechanism of EPM-GCNs and evaluate the models over various real-world network datasets. Empirical results show that the EPM-GCNs achieve state-of-the-art performance in most of the node and graph-level classification tasks.

## 2 EDGE PARTITION MODULATED GCNs

In this section, we present EPM-GCNs that learn the representation of node or graph from relation-specific sub-representations. As shown in Figure 1, the overall architecture of EPM-GCNs consists of a *generative network* (parameterized with $\boldsymbol{\theta}$) and an *inference network* (parameterized with $\boldsymbol{\omega}$). Section 2.1 outlines the data generation process and corresponding supportive modules in the *generative network*, Section 2.2 elaborates the module in the *inference network* as well as the inference-learning details.

### 2.1 GENERATIVE TASK-LEARNING WITH LATENT COMMUNITIES

For the generative supervised graph analytic tasks under discussion, the information of a graph $\mathcal{G}$ with $N$ nodes could be summarized by the triplet $(\mathbf{X}, \mathbf{A}, \mathbf{y})$, where $\mathbf{X}$ is the design matrix whose rows represent the features of individual nodes, $\mathbf{A}$ is the binary adjacency matrix of $\mathcal{G}$, with each "1" entry indicating the presence of an edge between the corresponding node pair, and $\mathbf{y} \in \{\mathbf{y}_o, \mathbf{y}_u\}$ is the set of labels, with subscripts $_o$ and $_u$ denoting the observed and unobserved parts of labels, respectively. The goal of our latent factor model is to describe the generation process of observed edges and labels. Based on that, we are enabled to predict the holdout labels from their posterior. The joint generation of edges $\mathbf{A}$ and labels $\mathbf{y}$ is modeled by stacking up the generation process of edges and the generation of labels given the edges.

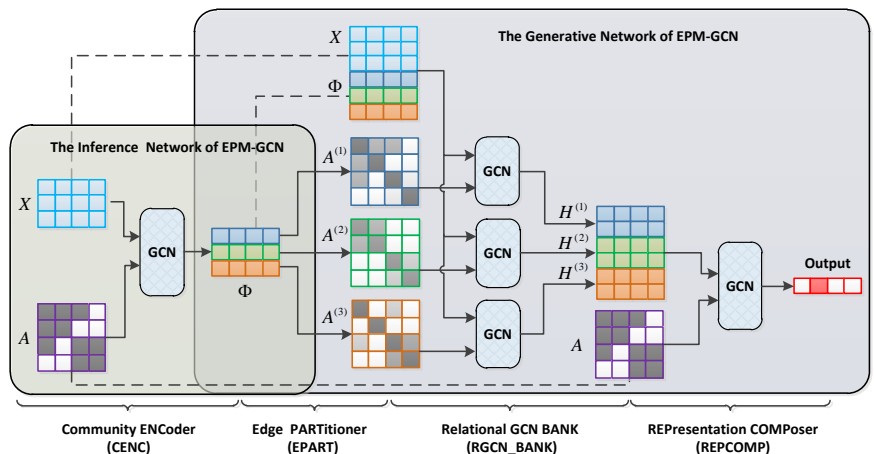

Figure 1: The overview of EPM-GCN's computation graph. $\boldsymbol{\Phi}$ encodes nodes' affiliation strength to each community, which is used to (1) enrich node features with community information (the dashed line) and (2) partition edges via normalized rank factorization (the solid arrows). They are then passed to an array of GCNs to generate node representations that correspond to heterogeneous community-based relations, and finally integrated together to optimize the classification task.

### 2.1.1 THE GENERATION OF EDGES

We adopt the generation process developed in Zhou (2015) to model the formation of edges $\mathbf{A}$:

$$\mathrm{A}_{i,j} \sim \mathrm{Ber}(p_{ij}), \ p_{ij} := 1 - e^{-\sum_{k=1}^{K} \gamma_k \Phi_{i,k} \Phi_{j,k}}, \tag{1}$$

which has an alternative representation under the Bernoulli-Poisson link as $\mathrm{A}_{i,j} = \mathbf{1}_{\mathrm{M}_{i,j} \geq 1}$, $\mathrm{M}_{i,j} \sim$ $\mathrm{Poisson}\big(\sum_{k=1}^{K} \gamma_k \Phi_{i,k} \Phi_{j,k}\big)$. In this expression, $\boldsymbol{\Phi}$ is a positive node-community affiliation matrix, whose entry at the $i$th row and $k$th column could be interpreted as the strength that node $i$ is affiliated with community $k$, $\gamma_k$ $(k \in [K])$ is a positive community activation level indicator that measures the member interaction frequencies featured by community $k$, and $\gamma_k \Phi_{i,k} \Phi_{j,k}$ quantifies the interaction rate between nodes $i$ and $j$ through community $k$. In our practice, we treat $\boldsymbol{\gamma} = [\gamma_1, \gamma_2, \cdots, \gamma_k]'$ as a part of the trainable parameters $\boldsymbol{\theta}$.

In this paper, we focus on another equivalent representation that partitions each edge into the logical disjunctions of $K$ latent binary edges, expressed as

$$\mathrm{A}_{i,j} = \vee_{k=1}^{K} \mathrm{A}_{i,j,k}, \ \mathrm{A}_{i,j,k} \sim \mathrm{Ber}\left(p_{ijk}\right), \ p_{ijk} := 1 - e^{-\gamma_k \Phi_{i,k} \Phi_{j,k}} \quad \forall i, j \in [N]^1$$

where $\vee$ is the logical disjunction (*i.e.*, logical OR) operator and $p_{ijk}$ denotes the probability that the interaction of nodes $i$ and $j$ in community $k$ results in an edge. The connection between nodes $i$ and $j$ would establish as long as their interaction in one community forms up an edge, in other words, nodes $i$ and $j$ would be disconnected only if their interactions in ALL communities fail to generate edges. Link function 1 gives the conditional probability $p_{\boldsymbol{\theta}}(\mathbf{A} \mid \boldsymbol{\Phi})$. Note that the node interaction over each community uniquely corresponds to one type of inter-node relation, the generation of $\mathbf{A}$ reflects the aggregation of these heterogeneous relations, suggesting that one would need to partition the edges to separate these relations.

### 2.1.2 THE GENERATION OF LABELS:

In consonance with the graph generative model that establishes the graph from latent communities, we develop the generation process of labels from all relation-specific representations. To achieve that, we first need to separate each relation from their totality, *i.e.*, factorize the edges with the *edge partitioner* module:

**Edge PARTitioner** (EPART). *The edge partitioner takes edges* $\mathbf{A}$ *and community-affiliation score matrix* $\boldsymbol{\Phi}$ *as inputs and returns* $K$ *positive-weighted edges:* $\{\mathbf{A}^{(1)}, \mathbf{A}^{(2)}, \cdots, \mathbf{A}^{(K)}\}$,

---

[1]We use the $[N]$ notation for the integer set $\{1, 2, \cdots, N\}$ throughout the text.

s.t. $\sum_{k=1}^{K} \mathbf{A}^{(k)} = \mathbf{A}$. *The partition function follows the form*

$$\mathbf{A}_{i,j}^{(k)} = \frac{e^{(\gamma_k \Phi_{i,k} \Phi_{j,k})/\tau}}{\sum_{k'} e^{(\gamma_{k'} \Phi_{i,k'} \Phi_{j,k'})/\tau}} = \frac{\left(\frac{1}{1-p_{ijk}}\right)^{\frac{1}{\tau}}}{\sum_{k'} \left(\frac{1}{1-p_{ijk'}}\right)^{\frac{1}{\tau}}}, \ k = 1, 2, \cdots, K, \ \forall i, j \in [N]. \quad (2)$$

Here $\tau$ denotes the "temperature" that controls the sharpness of weights, its value is specified in Table 6 in the appendix. The edge weights produced from *edge partitioner* represent the proportion that each community-based relation takes in contributing to edge formation. When $\tau$ is set sufficiently small, $\mathbf{A}^{(1)}, \mathbf{A}^{(2)}, \cdots, \mathbf{A}^{(K)}$ would approximate binary factors graphs of $\mathcal{G}$, we hence refer to them as "relational graph factors". Replacing $\gamma_k \Phi_{i,k} \Phi_{j,k}$ with $\mathbf{\Phi}_{i,k} \mathrm{diag}(\boldsymbol{\gamma}_k) \mathbf{\Phi}'_{j,k}$ generalizes Equation 2 to metacommunity-based edge partition, in this expression, the total number of communities is greater than $K$, and $\mathbf{\Phi}_{i,k}$ denotes the $k$th segment in node $i$'s community-affiliation encoding. The generalized *edge partitioner* enriches the choices of $\mathbf{\Phi}$'s dimensionalities, which enhances the flexibility in model implementation.

Subsequent to the *edge partitioner* follows the *relational GCN bank*:

**Relational GCN BANK** (RGCN_BANK). *The relational GCN bank is made up of $K$ independent GCN components, coupling with the corresponding relational graph factors. It takes $\mathbf{X}^* := \mathbf{X} \parallel \mathbf{\Phi}$ as input[2], and outputs* $\mathrm{g}_{\boldsymbol{\theta}}^{(1)}(\mathbf{X}^*), \mathrm{g}_{\boldsymbol{\theta}}^{(2)}(\mathbf{X}^*), \cdots, \mathrm{g}_{\boldsymbol{\theta}}^{(K)}(\mathbf{X}^*)$; $\mathrm{g}_{\boldsymbol{\theta}}^{(k)}(\cdot) := \mathrm{GCN}_{\boldsymbol{\theta}}(\cdot, \mathbf{A}^{(k)}), \ k \in [K]$.

The function of this design is to learn relation-wise separable node representations. We now qualitatively justify the fulfillment of this requirement. Without loss of generality, let us treat the relational graph factors as approximately binary, this means the each observed edge is annotated with the relation that makes the greatest contribution in its generation. Let us denote the relations as $r_1, r_2, \cdots, r_k$ which are uniquely featured with communities $c_1, c_2, \cdots, c_K$. Assuming node $u$ connects to its neighbors $v_1$ and $v_2$ by relations $r_1$ and $r_2$, respectively. Without edge partition, features of $v_1$ would be propagated to $v_2$ in two runs of neighborhood aggregation even if they connects to $u$ with different community identities. This issue is moderated by the *relational GCN bank*, as the aggregation path $(v_1, u)$ would only be sent to the GCN component for relation $r_1$ whereas the path $(u, v_2)$ would only exist in the GCN component corresponds to relation $r_2$. As a result, in each GCN component, nodes only share features through edges explained by the same relation, we hence consider the outputs of *relational GCN bank* as relation-specific.

Finally, we learn the representations of nodes from relation-specific sub-representations. This step involves the *representation composer* module:

**REPresentation COMPoser** (REPCOMP). *Let $\mathbf{H}^{(k)} := \mathrm{g}_{\boldsymbol{\theta}}^{(k)}(\mathbf{X}^*)$ denote the node representations learned from the $k$th relation, $k \in [K]$, and function $f(\cdot)$ denote the representation composer, whose functionality is to project a composite of relation-specific node representations to one representation matrix, i.e., $\mathbf{H}_{\mathcal{V}} = f\left(\mathbf{H}^{(1)}, \mathbf{H}^{(2)}, \cdots, \mathbf{H}^{(K)}\right) := \mathrm{GCN}_{\boldsymbol{\theta}}\left(\|_{k=1}^{K} \mathbf{H}^{(k)}, \mathbf{A}\right)$. We could further obtain $\mathbf{h}_{\mathcal{G}}$, the vector representation of graph $\mathcal{G}$, by sending the node level representations $\mathbf{H}_{\mathcal{V}}$ to a graph pooling layer, i.e., $\mathbf{h}_{\mathcal{G}} := \mathrm{GRAPHPOOL}(\mathbf{H}_{\mathcal{V}})$.*

Taking softmax on the feature dimension of the *representation composer*'s output gives the uncertainty distribution of labels, from which we are able to make predictions on the category of the object (node or graph). Cascading the *edge partitioner*, the *community-GCN bank* with the *representation composer* yields the probability $p_{\boldsymbol{\theta}}(\mathbf{y} \mid \mathbf{A}, \mathbf{\Phi}, \mathbf{X})$.

## 2.2 TRAINING EPM-GCN: INFERENCE AND LEARNING

### 2.2.1 THE REPARAMETERIZABLE EVIDENCE LOWER BOUND OBJECTIVE

We set the prior distribution of $\mathbf{\Phi}$ as $\Phi_{i,k} \overset{iid}{\sim} \mathrm{Gam}(\alpha, \beta), \forall (i, k) \in [N] \times [K]$, and use a *community encoder*[3] to approximate the posterior distribution $p(\mathbf{\Phi} \mid \mathbf{A}, \mathbf{y})$:

**Community ENCoder** (CENC). *The community encoder models the variational posterior $q_{\boldsymbol{\omega}}(\mathbf{\Phi})$ by a Weibull distribution with shape $\mathbf{K}$ and scale $\mathbf{\Lambda}$, both parameters are learned by a GCN, i.e.,*

---

[2]The operator $\parallel$ denotes "concatenation".

[3]The *community encoder* is the only module in the *inference network*.

$\mathbf{K} \parallel \boldsymbol{\Lambda} = \mathrm{GCN}_{\boldsymbol{\omega}}(\mathbf{X}, \mathbf{A})$. *A random sample from $q_{\boldsymbol{\omega}}(\boldsymbol{\Phi})$ could be created through the inverse CDF transformation of a uniformly distributed variable, given as follows:*

$$\boldsymbol{\Phi} = \boldsymbol{\Lambda} \odot \big(-\log(1 - \mathbf{U})\big)^{\circ(1 \oslash \mathbf{K})}, \ \ \mathrm{U}_{i,k} \overset{iid}{\sim} \mathrm{U}(0, 1), \ \forall (i, k) \in [N] \times [K].$$

*The operators $\odot$, $\oslash$ and $\circ$ (in the front of superscript[4]) denote element-wise multiplication, division and power, respectively.*

To jointly infer the latent factors $\boldsymbol{\Phi}$ and use them to learn the task, we optimize the evidence lower bound objective (ELBO) $\mathcal{L}$, given by Equation 3, *i.e.*,

$$\begin{aligned}
\mathcal{L} &= \mathbb{E}_{q_{\boldsymbol{\omega}}(\boldsymbol{\Phi})} \log p_{\boldsymbol{\theta}}(\mathbf{y}_o, \mathbf{A} \mid \boldsymbol{\Phi}, \mathbf{X}) - D_{\mathrm{KL}}(Q_{\boldsymbol{\Phi}} \| P_{\boldsymbol{\Phi}}) \\
&= \mathbb{E}_{q_{\boldsymbol{\omega}}(\boldsymbol{\Phi})} \log p_{\boldsymbol{\theta}}(\mathbf{y}_o \mid \mathbf{A}, \boldsymbol{\Phi}, \mathbf{X}) + \mathbb{E}_{q_{\boldsymbol{\omega}}(\boldsymbol{\Phi})} \log p_{\boldsymbol{\theta}}(\mathbf{A} \mid \boldsymbol{\Phi}) - D_{\mathrm{KL}}(Q_{\boldsymbol{\Phi}} \| P_{\boldsymbol{\Phi}}),
\end{aligned} \tag{3}$$

where the notations $P_{\boldsymbol{\Phi}}$ and $Q_{\boldsymbol{\Phi}}$ represent the prior and the variational posterior distributions of $\boldsymbol{\Phi}$, respectively. The three right hand side terms correspond to the classification task, graph reconstruction, and KL-regularization, we thereby refer to them as $\mathcal{L}_{\mathrm{task}}$, $\mathcal{L}_{\mathrm{rec}}$ and $\mathcal{L}_{\mathrm{KL}}$ in the sequel. Note that the aforementioned specifications of $\boldsymbol{\Phi}$'s prior and variational posterior, as in Zhang et al. (2018a), yield an analytical expression of $\mathcal{L}_{\mathrm{KL}}$.

### 2.2.2  THE OVERALL TRAINING ALGORITHM

To effectively train EPM-GCNs, we separate the overall training algorithm into an *unsupervised pretrain* stage, followed by a *supervised finetune* stage.

**The unsupervised pretrain:** The ultimate goal of the *community encoder* is to find a community assignment according to which edge partitions would facilitate task learning. Since the overall likelihood of node labels involves stacking multiple modules embodied by deep neural networks, inferring the posterior $p(\boldsymbol{\Phi} \mid \mathbf{A}, \mathbf{y}_o)$ is difficult from an optimization perspective. To seek better model convergence, we break this optimization problem into two stages, namely, we first train the *community encoder* to discover the latent communities that explains the graph architecture, then finetune the communities to align them to task-learning. The first stage is the *unsupervised pretrain*, at which we pretrain the *community encoder* until convergence by optimizing $\mathcal{L}_{\mathrm{rec}} + \mathcal{L}_{\mathrm{KL}}$, *i.e.*, updating model parameters $\boldsymbol{\omega}$ towards $\boldsymbol{\omega}^*$, which optimizes $\mathbb{E}_{q_{\boldsymbol{\omega}}(\boldsymbol{\Phi})} \log p_{\boldsymbol{\theta}}(\mathbf{A} \mid \boldsymbol{\Phi}) - D_{\mathrm{KL}}(Q_{\boldsymbol{\Phi}} \| P_{\boldsymbol{\Phi}})$. The variational probability $q_{\boldsymbol{\omega}}(\boldsymbol{\Phi})$ is learned to approximate $p(\boldsymbol{\Phi} \mid \mathbf{A})$. The heuristic that the *unsupervised pretrain* is generally supportive to the ultimate goal is that although communities and node categories describe node similarity from different aspects, these two aspects of similarity are semantically mutually informative, *i.e.*, the discovered communities from graph architecture is inherently correlated with node labels, which is suggested by previous work on graph representation learning with community information (Sun et al., 2019), in which the node classification results clearly benefit from the node representations that hold community information. Such assumption is further qualitatively verified in Section 4.2. We find through practice that the *unsupervised pretrain* not only stabilizes model convergence, but also provides a reasonable initial state for subsequent supervised optimization thus leads to better performances.

**The supervised finetune:** This stage involves iteratively running *inference* and *learning*. In the *inference* step, we fix the *generative network*, and train the *inference network* by optimizing the ELBO $\mathcal{L}$, given in Equation 3. Each *inference* step runs after $M$ *learning* steps, where we treat the *inference network* as stationary and update the *generative network* by optimizing $\mathcal{L}_{\mathrm{task}}$. Both $\mathcal{L}$ and $\mathcal{L}_{\mathrm{task}}$ are computed over the training set $\{\mathbf{y}_o, \mathbf{A}, \mathbf{X}\}$. The *supervised finetune* updates the approximation target of the *inference network* from $p(\boldsymbol{\Phi} \mid \mathbf{A})$ to $p(\boldsymbol{\Phi} \mid \mathbf{A}, \mathbf{y}_o)$, which finetunes community inference with label supervision, making the subsequent edge-partition more facilitative to the classification task.

The entire training pipeline of EPM-GCNs is summarized in Algorithm 1 (in Appendix A). After running the *unsupervised pretrain* and the *supervised fintune* stages, the variational distribution $q_{\boldsymbol{\omega}}(\boldsymbol{\Phi})$ is trained to approximate the posterior $p(\boldsymbol{\Phi} \mid \mathbf{y}_o, \mathbf{A})$. We hence approximate the predictive distribution of unobserved labels with $\mathbb{E}_{q_{\boldsymbol{\omega}}(\boldsymbol{\Phi})}[p_{\boldsymbol{\theta}}(\mathbf{y}_u \mid \mathbf{A}, \boldsymbol{\Phi}, \mathbf{X})]$, and estimate it via Monte Carlo estimation. In particular, we estimate $p(\mathbf{y}_u \mid \mathbf{A}, \mathbf{X}, \mathbf{y}_o)$ with $\frac{1}{S} \sum_{s=1}^{S} p_{\boldsymbol{\theta}}(\mathbf{y}_u \mid \mathbf{A}, \boldsymbol{\Phi}^{(s)}, \mathbf{X})$, where $\boldsymbol{\Phi}^{(s)} \overset{iid}{\sim} q_{\boldsymbol{\omega}}(\boldsymbol{\Phi})$ for $s = 1, \ldots, S$. For simplicity, we set $S = 1$ unless specified otherwise.

---

[4]Otherwise, the $\circ$ notation in the context of $f \circ g$ denotes function composition.

Table 1: Statistics of the datasets for node and graph-level classification..

| Task | Node Classification | | | Graph Classification | | | | | | | |
|---|---|---|---|---|---|---|---|---|---|---|---|
| Dataset | Cora | Citeseer | Pubmed | IMDB-B | IMDB-M | MUTAG | PTC | NCI1 | PROTEINS | RDT-B | RDT-M |
| Graphs | 1 | 1 | 1 | 1000 | 1500 | 188 | 344 | 4110 | 1113 | 2000 | 5000 |
| Edges | 5,429 | 4,732 | 44,338 | 96.5 | 65.9 | 19.8 | 26.0 | 32.3 | 72.8 | 497.7 | 594.8 |
| Features | 1,433 | 3,703 | 500 | 65 | 59 | 7 | 19 | 37 | 3 | 566 | 734 |
| Nodes | 2,708 | 3,327 | 19,717 | 19.8 | 13.0 | 17.9 | 25.5 | 29.8 | 39.1 | 29.6 | 508.5 |
| Classes | 7 | 6 | 3 | 2 | 3 | 2 | 2 | 2 | 2 | 2 | 5 |

## 3 RELATED WORK

**Deep graph generative models:** Advances in this line of work include adopting deep architectures in learning node embeddings (Kipf & Welling, 2016; Li et al., 2018) and joint optimization with associated graph-analytic tasks such as semi-supervised node classification (Hasanzadeh et al., 2019; Wang et al., 2020), community discovery (Sun et al., 2019; Mehta et al., 2019), *etc*. Similar to these models, we regularize the node embeddings $\mathbf{\Phi}$ by its probabilistic dependency with the graph structure, but instead of simply passing the learned embeddings to a black-box prediction network, we use them to develop the generative process of the labels, leading to better interpretability.

**GCNs modeling multi-relational data:** The general idea is to individually process node representations aggregated from different types of relations. This logistic is widely adopted by GCN variants handling graph data where relational annotations are either explicit (Schlichtkrull et al., 2018; Vashishth et al., 2020) or implicit. In the latter setup, the missing relations are conventionally completed with human knowledge (Abu-El-Haija et al., 2019; Vashishth et al., 2020), or learned via label supervision (Veličković et al., 2018; Gao & Ji, 2019; Ma et al., 2019; Yang et al., 2020). Contrary to these methods which treat edges as given, we jointly model the generation of labels as well as the edges, and systematically model the relations by posterior inference. The inferred relations are then used to partition edges, learn relation-specific sub-representations, and finally optimize the downstream tasks.

## 4 EMPIRICAL EVALUATION

We compare the proposed EPM-GCNs against related baselines on two fundamental tasks: node classification and graph classification. In the sequel, we use the suffixes -n and -g to distinguish the variants of EPM-GCN developed for node and graph-level classification tasks, respectively.

### 4.1 DATASETS PREPARATION

We evaluate EPM-GCNs on 11 benchmarks. For node classification, we use 3 citation networks: Cora, Citeseer and Pubmed. All three datasets provide bag-of-words document representations as node features and (undirected) citations as edges. For graph classification, we use 4 bioinformatics datasets (MUTAG, PTC, NCI1, PROTEINS) and 4 social network datasets (IMDB-BINARY, IMDB-MULTI, REDDIT-BINARY and REDDIT-MULTI). The input node features are crafted in the same way with Xu et al. (2019). We summarize the statistcis of selected datasets in Table 1.

### 4.2 VISUALIZING LEARNED REPRESENTATIONS

To qualitatively assess the proposed model, we visualize the spatial distribution of $\mathbf{\Phi}$ obtained at the end of the *unsupervised pretrain* stage (Figures 2(a), 2(b)), and the relation-specific sub-representations obtained at the end of the *supervised finetune* stage (Figures 2(c), 2(d)). Node representation are projected to 2-D space via t-SNE (Van der Maaten & Hinton, 2008). We select Cora and MUTAG as representatives of the node and graph classification datasets. For the MUTAG dataset, we remove less than 10 graphs that contain node categories with less than 5 instances and randomly sample 10 graphs for visualization. In Figures 2(a) and 2(b), the spatial clustering of nodes from the same class indicates strong correlation between inferred latent communities and node labels, which not only demonstrates our model's ability in capturing community structure (even in small graphs with only tens of nodes), but also positively supports our model designs, including (i) input feature augmentation: $\mathbf{X}^* = \mathbf{X} \parallel \mathbf{\Phi}$ and (ii) edge partition. Figures 2(c) and 2(d) further justifies that the sub-representations $\{g_{\boldsymbol{\theta}}^{(1)}(\mathbf{X}^*), g_{\boldsymbol{\theta}}^{(2)}(\mathbf{X}^*), \cdots, g_{\boldsymbol{\theta}}^{(K)}(\mathbf{X}^*)\}$ are not repetitive, signaling that the proposed model successfully extracts relation-specific representations, which potentially enhance

the discriminative power of node or graph representations and lead to better performances on the downstream tasks.

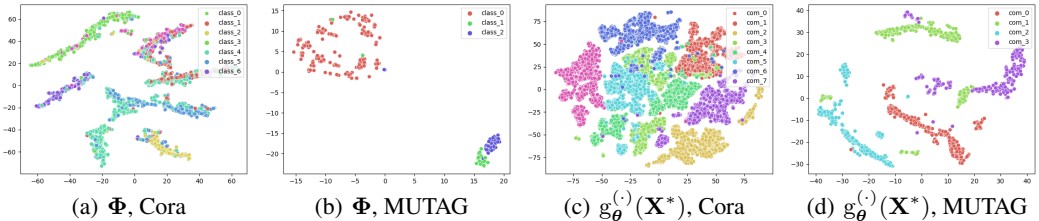

(a) $\Phi$, Cora      (b) $\Phi$, MUTAG      (c) $g_{\theta}^{(\cdot)}(\mathbf{X}^*)$, Cora      (d) $g_{\theta}^{(\cdot)}(\mathbf{X}^*)$, MUTAG

Figure 2: Visualization of the spatial distribution of node representations learned by EPM-GCNs on Cora and MUTAG datasets, mapped to 2-D space. (a) and (b) plot the t-SNE projections of nodes' community-affiliation scores (*i.e.*, rows in $\Phi$) at the end of the *unsupervised pretrain* stage, where nodes are colored by their categories. (c) and (d) plot the t-SNE projections of relation-specific sub-representations of all nodes over all types of relations obtained at the end of the *supervised finetune* stage, the colors annotate outputs from different GCN components in the *relational GCN bank* (*i.e.*, each color corresponds to a relation, each node corresponds to $K$ points in different colors). Spatially clustering of each colored group supports our community-based edge partition and node feature augmentation ((a), (b)), and justifies the learned relation-specific sub-representations are indeed relation-wise separable.

## 4.3 NODE CLASSIFICATION

**Experimental settings:** To make a fair comparison, we use the train/test/validation split standardized by the vanilla GCN (Kipf & Welling, 2017) for Cora, Citeseer and Pubmed datasets. For the *community encoder*, we adopt a similar encoder structure devised by Kipf & Welling (2016) with softplus activations, which outputs 16-dimensional community-affiliation scores for each node. We set the number of metacommunities $K = 4$ for Cora and Citeseer and $K = 8$ for Pubmed dataset, the values of $K$ are selected through cross-validation. The relational graph factors are then obtained through the generalized edge partition. Before being sent to the *relational GCN bank*, these graph factors are row-normalized by sparse softmax. We implement the *relational GCN bank* with $K$ single-layer GCNs with ReLU activation, following (Veličković et al., 2018), we set their output dimensions as $64/K$. The *representation composer* is also implemented with a single-layer GCN. With these specifications, the amount of parameters in the *generative network* remains comparable with that of a 2-layer GCN with 64 hidden units (GCN-64) regardless of the choice of $K$. We record the complete hyperparameter specifications in Table 6 (in Appendix C).

**Performance comparison:** We use classification accuracy as the evaluation metric in the node classification tasks. Table 2 reports EPM-GCN-*n*'s average performances ($\pm$ standard error) against the paper records of the average performances of 3 groups of related baselines on the three citation network datasets, where * marks the results reported by Veličković et al. (2018). The first group

Table 2: Comparisons on node classification performances.

| Method | Cora | Citeseer | Pubmed |
|---|---|---|---|
| ChebNet (Defferrard et al., 2016) | 81.2 | 69.8 | 74.4 |
| GCN (Kipf & Welling, 2017) | 81.5 | 70.3 | 79.0 |
| GCN-64* | 81.4 | 70.9 | 79.0 |
| SIG-VAE (Hasanzadeh et al., 2019) | 79.7 | 70.4 | 79.3 |
| WGCAE (Wang et al., 2020) | 82.0 | 72.1 | 79.1 |
| GAT* (Veličković et al., 2018) | 83.0 | 72.5 | 79.0 |
| hGAO (Gao & Ji, 2019) | 83.5 | 72.7 | 79.2 |
| DisenGCN (Ma et al., 2019) | 83.7 | **73.4** | 80.5 |
| EPM-GCN-*n* (this work) | **84.0** $\pm$ 0.1 | 72.4 $\pm$ 0.1 | **82.2** $\pm$ 0.2 |

(Defferrard et al., 2016; Kipf & Welling, 2017) are plain GCNs that aggregated node features with only one set of weighted edges. These models are designed to handle at most two types of node relations: connected or not, so it is not surprised to find that EPM-GCN-*n* outperforms these models by a wide margin. Since removing (i) relational inference and (ii) relation-specific feature learning degenerates EPM-GCN-*n* to GCN-64, the improvement of our model against GCN-64 could be explained as the joint effects of the mentioned two practices. The second group (Hasanzadeh et al., 2019; Wang et al., 2020) are graph generative models that jointly optimized with semi-supervised loss. Both SIG-VAE (Hasanzadeh et al., 2019) and WGCAE (Wang et al., 2020) adopt similar graph generative model and loss structure to ours, the difference is that these models directly projects the latent representations to node labels by a deterministic prediction network, whereas we sample labels from a edge-partition based label generation process. Therefore our improvement with respect to the second group could be attributed to the marginal benefit from our label generative model. The third group of models (Veličković et al., 2018; Gao & Ji, 2019; Ma et al., 2019) employ multi-head

Table 3: Comparisons on graph classification performances.

| Method | IMDB-B | IMDB-M | MUTAG | PTC | NCI1 | PROTEINS | RDT-B | RDT-M |
|---|---|---|---|---|---|---|---|---|
| WL subtree (Shervashidze et al., 2011) | 73.8 ± 3.9 | 50.9 ± 3.8 | 90.4 ± 5.7 | 59.9 ± 4.3 | 86.0 ± 1.8 | 75.0 ± 3.1 | 81.0 ± 3.1 | 52.5 ± 2.1 |
| AWL (Ivanov & Burnaev, 2018) | 74.5 ± 5.9 | 51.5 ± 3.6 | 87.9 ± 9.8 | - | - | - | 87.9 ± 2.5 | 54.7 ± 2.9 |
| DCNN (Atwood & Towsley, 2016) | 49.1 ± 0.0 | 33.5 ± 0.0 | 67.0 ± 0.0 | 56.6 ± 0.0 | 62.6 ± 0.0 | 61.3 ± 0.0 | - | - |
| DGCNN (Zhang et al., 2018b) | 70.0 ± 0.0 | 47.8 ± 0.0 | 85.8 ± 0.0 | 58.6 ± 0.0 | 74.4 ± 0.0 | 75.5 ± 0.0 | - | - |
| PATCHYSAN (Niepert et al., 2016) | 71.0 ± 2.2 | 45.2 ± 2.8 | 92.6 ± 4.2 | 60.0 ± 4.8 | 78.6 ± 1.9 | 75.9 ± 2.8 | 86.3 ± 1.6 | 49.1 ± 0.7 |
| GIN-0 (Xu et al., 2019) | 75.1 ± 5.1 | 52.3 ±2.8 | 89.4 ±5.6 | 64.6 ± 7.0 | 82.7 ± 1.7 | 76.2 ± 2.8 | 92.4 ± 2.5 | 57.5 ± 1.5 |
| GIN-$\epsilon$ (Xu et al., 2019) | 74.3 ± 5.1 | 52.1 ± 3.6 | 89.0 ± 6.0 | 63.7 ± 8.2 | 82.7 ± 1.6 | 75.9 ± 3.8 | 92.2 ± 2.3 | 57.0 ± 1.7 |
| FactorGCN (Yang et al., 2020) | 75.3 ± 2.7 | - | 89.9 ± 6.5 | - | - | - | - | - |
| GCN (Kipf & Welling, 2017) | 74.0 ± 3.4 | 51.9 ± 3.8 | 85.6 ± 5.8 | 64.2 ± 4.3 | 80.2 ± 2.0 | 76.0 ± 3.2 | 50.0 ± 0.0 | 20.0 ± 0.0 |
| GAT (Veličković et al., 2018) | 70.5 ± 2.3 | 47.8 ± 3.1 | 89.4 ± 6.1 | 66.7 ± 5.1 | - | - | - | - |
| GraphSAGE (Hamilton et al., 2017) | 72.3 ± 5.3 | 50.9 ± 2.2 | 85.1 ± 7.6 | 63.9 ± 7.7 | 77.7 ± 1.5 | 75.9 ± 3.2 | - | - |
| EPM-GCN-$g$ (this work) | **76.7 ± 3.1** | **54.1 ± 2.1** | **93.6 ± 3.4** | **75.6 ± 5.9** | 83.9 ± 1.8 | **80.5 ± 2.8** | 90.5 ± 1.8 | 55.0 ± 1.5 |

Table 4: Comparisons of EPM-GCN-$g$ with various input node features.

| Random | Hand-crafted | Community-based | IMDB-B | IMDB-M | MUTAG | PTC |
|---|---|---|---|---|---|---|
| ✓ | | | 64.7 ± 1.6 | 42.3 ± 1.5 | 84.6 ± 4.3 | 63.6 ± 2.0 |
| | ✓ | | **80.3 ± 2.0** | 53.5 ± 2.6 | 93.1 ± 5.0 | 74.7 ± 4.1 |
| | | ✓ | 74.7 ± 5.1 | 51.5 ± 2.0 | 88.9 ± 5.5 | 68.9 ± 3.9 |
| | ✓ | ✓ | 76.7 ± 3.1 | **54.1 ± 2.1** | **93.6 ± 3.5** | **75.6 ± 5.9** |

and adaptively weighted feature aggregation mechanisms, which greatly enhances their ability in handling multi-relational data. However, these models lack systematic modeling of latent relations, which on the other hand is the strength of our model. Our performance boost against the third group on most of the benchmarks demonstrates the marginal benefit of our relational inference model.

## 4.4 GRAPH CLASSIFICATION

**Experimental settings:** For EPM-GCN-$g$, we implement the *community encoder* with a 2-layer GCN and produces the latent community-affiliation encoding with 100 dimensions. The number of metacommunities is set to $K = 4$ and corresponding relational graph factors are generated through the generalized edge partition. We choose GIN layer as the building block to define the rest of the model, and set the total network depth as 5 (including the input layer) such that EPM-GCN-$g$ is comparable with the vanilla GIN (Xu et al., 2019). We define the $K$ components in the *relational GCN bank* by sharing the input layer and evenly partitioning the neuron units in the 2 top hidden layers. The 2 ending layers are treated as the *representation composer*. The remaining implementation details are included in Table 6 (in Appendix C).

**Performance comparison:** We follow GIN (Xu et al., 2019) to perform 10-fold cross validation for each dataset and report the average classification accuracy ($\pm$ standard error). As shown in Table 3, we compare the developed EPM-GCN-$g$ with several state-of-the-art baselines for graph classification, including the WL subtree kernel (Shervashidze et al., 2011), Anonymous Walk Embeddings (AWL) (Ivanov & Burnaev, 2018) and other deep learning architectures like DCNN (Atwood & Towsley, 2016), DGCNN (Zhang et al., 2018b), and PYTCHY-SAN (Niepert et al., 2016). We also compare EPM-GCN-$g$ with five supervised GNNs including GCN (Kipf & Welling, 2017), GAT (Veličković et al., 2018), GraphSAGE (Hamilton et al., 2017), and two variants of GIN (Xu et al., 2019): GIN-0 and GIN-$\epsilon$. Thanks to our uniquely designed multi-relational inference as well as relation-specific graph representation learning and composition, EPM-GCN-$g$ achieves state-of-the-art graph classification performances on 5 out of 8 benchmarks, and the second-best performance on NCI1 dataset, where it outperforms all the other deep architectures. Notably, EPM-GCN-$g$ achieves 75.6% accuracy on PTC dataset with a significant 7.9% improvement over the second place.

## 4.5 ABLATION STUDY

To further study the marginal benefit of each practice we take in implementing EPM-GCN, as well as model's overall sensitivity to variations in hyperparameter setup, we evaluate EPM-GCN's graph classification performances over various choices of input features, number of metacommunities, and network structure.

**Various input features:** We first evaluate model performances with four types of inputs: random noise, hand-crafted features by Xu et al. (2019), community-affiliation scores, and hand-crafted features concatenated with community-affiliation scores. Results are shown in Table 4. The classification accuracy measures the discriminative level of graph representations learned from these four types of inputs, comparing the first and second rows with the third row tells us that community-affiliation scores are meaningful node representations and comparably informative to the task as

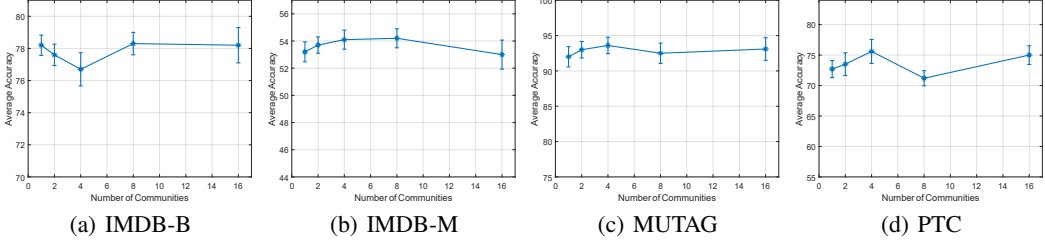

|  |  |  |  |
| :---: | :---: | :---: | :---: |
| (a) IMDB-B | (b) IMDB-M | (c) MUTAG | (d) PTC |

Figure 3: The average graph classification accuracy of EPM-GCN-$g$ with various settings on the numer of metacommunities $K \in \{1, 2, 4, 8, 16\}$. Results on (a) IMDB-BINARY, (b) IMDB-MULTI, (c) MUTAG, and (d) PTC datasets are shown in the subfigures.

hand-crafted node features. We could further conclude by comparing the second row with the fourth row that the information $\Phi$ and $X$ hold for the task enhances the discriminative level of learned representations in a complementary manner. Both conclusions justify the quality of $\Phi$ as additional node features.

**Number of communities:** We next study the variation of the model's performances with growing number of metacommunities. Figure 3 plots the average graph classification accuracy over 10-fold cross-validation against different selections of $K \in \{1, 2, 4, 8, 16\}$. In the context of graph classification, $K$ denotes the number of all metacommunities with at least one presence among the graphs in the dataset. We could see that the optimal results are typically achieved with $K > 1$, which is in consistent with the model assumption that the graph contains multiple underlying relations, the result also reflects the efficacy of the proposed model in modeling heterogeneous latent relations.

**Network structures:** Among our proposed results in Table 3, the network structure for the label generation is not particularly fine-selected and hidden layers are evenly distributed to the *relational GCN bank* and the *representation composer*, In this part, we focus on studying how the balance between these two modules influences the model performance on downstream tasks. We use the name convention EPM-GCN-$g$-$\{L_1\}$-$\{L_2\}$ to denote the variant with $L_1$ layers (including the input layer) in the *relational GCN bank* and $L_2$ layers in the *representation composer*, where $L_1 > 0$ and $L_1 + L_2 = 5$. As Table 5 shows, although the optimal layer assignments for each dataset are not the same, *i.e.*, the IMDB-B and IMDB-M datasets generally require less relation-wise locality in node representations than MUTAG or PTC datasets, both modules are verified to be necessary for the graph-analytic tasks, because neither EPM-GCN-$g$-$\{1\}$-$\{4\}$ nor EPM-GCN-$g$-$\{5\}$-$\{0\}$ triumphs over other layer combinations in any of these experiments. Additionally, the variation of model performances under different layer assignment demonstrates that datasets with larger amount of graphs are generally more insensitive to the change of model architecture, so the way to increase model robustness on datasets with less graphs would be a topic worth studying in the future.

Table 5: Comparisons of EPM-GCN-$g$ with various network structures.

| Network Structures | IMDB-B | IMDB-M | MUTAG | PTC |
| :--- | :---: | :---: | :---: | :---: |
| GIN-0 (Xu et al., 2019) | $75.1 \pm 5.1$ | $52.3 \pm 2.8$ | $89.4 \pm 5.6$ | $64.6 \pm 7.0$ |
| GIN-$\epsilon$ (Xu et al., 2019) | $74.3 \pm 5.1$ | $52.1 \pm 3.6$ | $89.0 \pm 6.0$ | $63.7 \pm 8.2$ |
| EPM-GCN-$g$-$\{1\}$-$\{4\}$ | $75.6 \pm 3.6$ | $55.4 \pm 2.7$ | $89.8 \pm 7.6$ | $70.9 \pm 5.3$ |
| EPM-GCN-$g$-$\{2\}$-$\{3\}$ | $\mathbf{77.8} \pm 2.0$ | $\mathbf{56.3} \pm 2.1$ | $91.0 \pm 4.9$ | $73.3 \pm 3.6$ |
| EPM-GCN-$g$-$\{3\}$-$\{2\}$ | $76.7 \pm 3.1$ | $54.1 \pm 2.1$ | $\mathbf{93.6} \pm 3.5$ | $\mathbf{75.6} \pm 5.9$ |
| EPM-GCN-$g$-$\{4\}$-$\{1\}$ | $74.5 \pm 2.2$ | $54.3 \pm 3.2$ | $87.0 \pm 6.8$ | $66.3 \pm 5.0$ |
| EPM-GCN-$g$-$\{5\}$-$\{0\}$ | $74.2 \pm 2.9$ | $51.2 \pm 2.5$ | $80.4 \pm 10.4$ | $63.1 \pm 6.2$ |

## 5 CONCLUSION

Moving beyond treating the graph adjacency matrix as given, we develop novel EPM-GCNs to aggregate the node interactions over multiple overlapping node communities. More specifically, we first construct a *community encoder* to project each node to its community-affiliation scores, and then partition the edges between nodes according to the edge-formation contribution of each community-based relation with a *edge partitioner*. Then we obtain the relation-specific sub-representations with the *relation GCN bank* and finally put them together to make predictions with a *representation composer*. Extensive qualitative and quantitative experiments on both node and graph-level has been made to demonstrate the efficacy of our model.

REPRODUCIBILITY STATEMENT

The PyTorch code to reproduce the results in the paper has been included in the supplementary material. Detailed hyperparameter settings have been provided in Table 6 in the Appendix.

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

APPENDIX

## A  THE PSEUDOCODE OF TRAINING ALGORITHM

---

**Algorithm 1:** The overall training algorithm of EPM-GCN

---
**Data:** node features $\mathbf{X}$, observed edges $\mathbf{A}$ and observed labels $\mathbf{y}_o$
**Modules**  : CENC, EPART, RGCN_BANK and REPCOMP
**Parameters:** $\boldsymbol{\omega}$ in the *inference network*, and $\boldsymbol{\theta}$ in the *generative network*
Initialize $\boldsymbol{\omega}$ and $\boldsymbol{\theta}$;
**while** *the community encoder is not converged* **do**
  |  $\boldsymbol{\Phi}, \mathbf{K}, \boldsymbol{\Lambda} \leftarrow \text{CENC}(\mathbf{A}, \mathbf{X})$;
  |  Compute $\mathcal{L}_{\text{rec}}$ and $\mathcal{L}_{\text{KL}}$, given in Equation 3;
  |  $\boldsymbol{\omega} \leftarrow \boldsymbol{\omega} - \eta_{\text{unsup}} \cdot \nabla_{\boldsymbol{\omega}}(\mathcal{L}_{\text{rec}} + \mathcal{L}_{\text{KL}})$;     /* *Unsupervised pretrain* */
**end**
**while** *the whole model is not converged* **do**
  |  $\boldsymbol{\Phi}, \mathbf{K}, \boldsymbol{\Lambda} \leftarrow \text{CENC}(\mathbf{A}, \mathbf{X})$;
  |  $\mathbf{A}^{(1)}, \mathbf{A}^{(2)}, \cdots, \mathbf{A}^{(K)} \leftarrow \text{EPART}(\boldsymbol{\Phi}, \mathbf{A})$;
  |  **for** *step* $\leftarrow 1$ **to** $M$ **do**
  |  |  $\hat{p}_{\mathbf{y}_o} \leftarrow \text{softmax}\big(\text{REPCOMP} \circ \text{RGCN\_BANK}(\boldsymbol{\Phi}, \mathbf{X}, \mathbf{A}^{(1)}, \mathbf{A}^{(2)}, \cdots, \mathbf{A}^{(K)})\big)$;
  |  |  Compute $\mathcal{L}_{\text{task}}$, given in Equation 3;
  |  |  $\boldsymbol{\theta} \leftarrow \boldsymbol{\theta} - \eta_{\text{sup},\boldsymbol{\theta}} \cdot \nabla_{\boldsymbol{\theta}}\mathcal{L}_{\text{task}}$;     /* The *learning* step */
  |  **end**
  |  Compute $\mathcal{L}$;
  |  $\boldsymbol{\omega} \leftarrow \boldsymbol{\omega} - \eta_{\text{sup},\boldsymbol{\omega}} \cdot \nabla_{\boldsymbol{\omega}}\mathcal{L}$;     /* The *inference* step */
**end**

---

## B  PROPERTY OF WEIBULL DISTRIBUTION

### • Similar PDF with Gamma Distribution

The Weibull distribution owns similar probability density functions (PDF) with a gamma one, which makes it flexible to model sparse and nonnegative latent representations:

$$\text{Weibull PDF: } P(x|k, \lambda) = \frac{k}{\lambda^k} x^{k-1} e^{(x/\lambda)^k},$$
$$\text{Gamma PDF: } P(x|\alpha, \beta) = \frac{\beta^\alpha}{\Gamma(\alpha)} x^{\alpha-1} e^{-\beta x}. \tag{4}$$

### • Easily Reparameterization

The latent variable $x \sim \text{Weibull}(k, \lambda)$ can be easily reparameterized as

$$x = \lambda(-\ln(1 - \varepsilon))^{1/k}, \ \ \varepsilon \sim \text{Uniform}(0, 1), \tag{5}$$

leading to a similar gradient calculation with the Gaussian reparameterization.

### • Analytic KL-Divergence

Moreover, the KL-divergence between the Weibull and gamma distributions has an analytic expression formulated as

$$\text{KL}(\text{Weibull}(k, \lambda)||\text{Gamma}(\alpha, \beta)) = -\alpha \ln \lambda + \frac{\gamma \alpha}{k}$$
$$+ \ln k + \beta \lambda \Gamma(1 + \frac{1}{k}) - \gamma - 1 - \alpha \ln \beta + \ln \Gamma(\alpha). \tag{6}$$

## C  IMPLEMETATION DETAILS

In our experiments, we find the initial state of inference module provided by the *unsupervised pretrain* stage is essential to EPM-GCN's performances.

The hyperparameters are either inherited from base models or selected on the basis of cross-validation. Note that for the graph-level classification task, the parameters are shared across experiments on all datasets. And the reported results are potentially improvable if dataset-specific cross-validations are applied.

Table 6: Hyperparameters settings for EPM-GCN.

| Hyperparameters | Experiments | |
|---|---|---|
| | Node Classification | Graph Classification |
| **Community ENCoder** | Settings | |
| $\alpha$ | 1 | 1 |
| $\beta$ | 1 | 1 |
| epoches of *unsupervised pretrain* | 1500 | 1500 |
| learning rate of *unsupervised pretrain* | 1e-3 | 1e-2 |
| batch size of *unsupervised pretrain* | 1 | 32 |
| type of GNN layers | GCN | GCN |
| module structure | {32}-{16} | {200}-{100} |
| **Edge PARTitioner** | Settings | |
| number of communities | [4,8] | 4 |
| the temperature $\tau$ for partition | 1 | 1 |
| **Relational GCN BANK** | Settings | |
| epoches of jointly training | 200 | 100 |
| learning rate of jointly training | 1e-2 | 1e-3 |
| weight decay of jointly training | 5e-4 | 0.0 |
| batch size of jointly training | 1 | 32 |
| concat weight of $\Phi$ | 3e-2 | 3e-4 |
| type of GNN layers | GCN | GIN |
| module structure | {64} | {64}-{64} |
| **REPresentation COMPoser** | Settings | |
| epoches of jointly training | 200 | 100 |
| learning rate of jointly training | 1e-2 | 1e-3 |
| weight decay of jointly training | 5e-4 | 0.0 |
| batch size of jointly training | 1 | 32 |
| type of GNN layers | GCN | GIN |
| module structure | {class num} | {64}-{64}-{class num} |

