# OpenReview forum: "Edge Partition Modulated Graph Convolutional Networks"
_ICLR.cc/2022/Conference — ICLR 2022 Submitted_

### Official Review · Reviewer_1iDH · 2021-10-24

**Correctness:** 2
**Technical Novelty And Significance:** 2
**Empirical Novelty And Significance:** 2
**Recommendation:** 3
**Confidence:** 5

**Main Review:**

Apart from statements that need clarification or adjustments, the paper is generally well presented and organized. Technically speaking, there are three main issues with the submission that can be summarized by: i) lack of proper motivations for the proposed architecture; ii) missing or ambiguous technical details; iii) robustness of the empirical evaluation. This review will proceed by sections, by listing major and minor comments to provide a feedback as complete as possible.

### Abstract

#### Major

From "In this paper" to "OR mechanism", the text is too dense and does not help the reader grasp the intuition of what is happening. Also, it is not clear why modeling latent node communities should be important to solve the main supervised task (point i) above).

#### Minor

"have demonstrated the working mechanisms" should be removed, as the paper does not fulfil the promise made by the authors.

### Introduction

#### Major

When referring to GCNs works, the authors did not mention two of the earliest works in the graph learning literature, namely GNN (Scarselli et al, 2009) and NN4G (Micheli, 2009). It is recommended that at least these two works are added to the paper.

From "the absence of relational inference" to the end of the paragraph: this statement looks like pure speculation unless supported by scientific evidence or concrete examples.

The paper seems to lack proper motivations, apart from the "need of modeling latent communities". Since the empirical evaluation relies on numeric results and a qualitative visualization, it is unclear why one should really need to model latent communities by using multiple GCNs. It is strongly recommended that the authors provide convincing motivations and justification for the development of their approach, other than stating that "it performs better" (see my comments below on the empirical evaluation).

#### Minor

Q: why is the GAT aggregation too "ad-hoc"? It seems pretty general and completely adaptive. And what does "ad-hoc" refers to?

Q: could the authors please describe how and why "extra information provided with the observed edges remedies optimization issues caused by the scarcity of label supervision"? Otherwise this seems another claim unsupported by evidence (either from the paper or from the literature).

Q: what are "observed (target) labels"? Are we considering a specific kind of task that considers unobserved and observed targets, like the semi-supervised setting of node classification? If that is the case, the authors should specify it, since that statement does not hold in general.

Please avoid using terms like "insatiable".

### Methodology

#### Major

The use of the terms "generative" and "inference" in contraposition can create confusion among readers. One of the reasons is that  the term "generative model" is often synonym of probabilistic models that perform inference, so "reconstruction" or "community encoder" could be more appropriate and disambiguating here. This distinction seems to confuse the authors as well: the "generative" model is associated with parameters \theta at the beginning of Section 2, but it seems that it is the other way round, with the parameters of the community encoder being \omega.
It would also benefit the presentation to state what is the role of the generative and inference models, to have a clearer picture in mind of what is to come.

Notation-wise, at this stage it remains unclear why the distinction between y_o and y_u has to be made, and the symbols representing the original and reconstructed adjacency matrix coincide.

The last statement of Sec 2.1.1 is crucial to the justification of your work, but it is again unsupported by scientific evidence.

Throughout the full text, there seems to be no reference whatsoever to the temperature values tried in the experiments. Please specify them.

The manuscript would probably benefit from a descriptive and empirical comparison with respect to Relational GCN (Schlichtkrull et al 2018), to understand why and whether or not your inductive bias is really more effective. For instance, R-GCN could be applied to the different adjacency matrices identified by the community encoder, as if edges had different types.

There is not proof or derivations for the equations of the community encoder in Sec 2.2.1, and the analytical expression of L_KL should be described somewhere.

From "and the model design that involves... to the end of the paragraph": this is another claim unsupported by scientific evidence, and it is even not very clear. It could be better to drop it or add some support to it.

Since the model is end-to-end trainable, it is not completely intuitive why the supervised finetuning stage should alternate between two different supervised optimizations, one of which is included into the other. The suspect is that this is a choice driven by necessity of convergence than to make the edge partition "more facilitative to the classification task". It would be appropriate to conduct an ablation study in which the model is finetuned in an end-to-end fashion, to show that this peculiar training strategy is indeed effective. Note that, at this stage of the article, it is still unclear what is the purpose of y_o and y_u.

Setting S=1 cannot be justified by mere simplicity. The value of S has an important effect on the quality of the approximation, so it would be appropriate to rephrase that or conduct a study that shows how S=1 is indeed sufficient to provide good performances.

#### Minor

Q: what is the meaning of "unsupervised parts of labels"?

It would help the discussion to see an actual example of community-specific adjacency matrices, as well as computing a discrepancy score between them.

Please refer to equations instead of "link functions"

"K positive-weighted edges" --> "K positive-weighted matrices"

"edge partitioner represents" --> represent refers to "edge weights", please correct these severe mistakes across the manuscript.

Could the authors please define "underqualified" and "data-label ratio"? This would help the comprehension of the manuscript.

### Empirical Evaluation

#### Major

All empirical evaluations in the paper follow protocols that have been heavily criticized by the community [1,2], either because they  are unfair, not robust, optimized wrt the test set, or all of the previous. The node classification tasks should have averaged the results over different train/val/test splits, and the 10-fold cv procedure of Xu et al. is known to optimize hyper-parameters on the test set. For these reasons, the empirical results in this paper should not be considered of sufficient quality, and the authors should set up a proper empirical evaluation. This includes:

1) Being clear in separating model selection from model assessment, as well as stating the protocols used in the paper
2) Testing a sensible amount of hyper-parameters configurations during model selection, rather than fixing most of them like in this paper. This is a computationally heavy phase, but it cannot be avoided in general.
3) Conducting ablation studies and hyper-parameters sensitivity tests as part of the output produced during model selection (on the validation set). In this paper, it is evident that the results of Table 3 are the result of the analysis of different configurations of Table 5, meaning the hyper-parameters have been optimized on the test set (see MUTAG and PTC, for example). The effect of hyper-parameters should never be studied on the test set.

[1] Shchur, Oleksandr, et al. "Pitfalls of graph neural network evaluation." arXiv preprint arXiv:1811.05868 (2018).

[2] Errica, Federico, et al. "A fair comparison of graph neural networks for graph classification." ICLR 2020.


**Summary Of The Paper:**

This article proposes a new architecture to solve node and graph classification tasks. The main idea is that an initial transformation is used to map nodes to K different communities, after which community-independent node features are computed and used together to classify the nodes/graph. The training process undergoes an initial unsupervised pre-training stage, followed by a supervised finetuning stage that, for some reason not well understood, alternates the optimization of two supervised objectives (one of which is included in the other), by fixing different parts of the end-to-end architecture at each step.


**Summary Of The Review:**

In light of the above considerations, the paper needs substantial rewriting and the empirical evaluations must be done following a proper protocol. It cannot be accepted as is, and probably there is no time to properly fix these points in the rebuttal phase.

---

> ### Author Response · Authors · 2021-11-22
> **We clarify the reviewer's questions and concerns, and address our concern for some disputable points made in the review**
>
> $\newcommand{\Phimat}{\boldsymbol{\Phi}}$
> $\newcommand{\Amat}{\mathbf{A}}$
> $\newcommand{\Xmat}{\mathbf{X}}$
> $\newcommand{\yv}{\boldsymbol{y}}$
>
> We appreciate the detailed comments and constructive suggestions from reviewer 1iDH. We have carefully revised the manuscript to address your comments. We will also add the ablation study of unsupervised pretrain (Please refer to our general comment for some examples of the impact of pretraining stage).
>
> **Abstract**
>
> > From "In this paper" to "OR mechanism", the text is too dense and does not help the reader grasp the intuition of what is happening. Also, it is not clear why modeling latent node communities should be important to solve the main supervised task (point i) above).
>
> We have revised our abstract to add more description and better highlight our motivations. Specifically, a rich set of prior works has illustrated the existence of latent node communities.  While such latent structure has been widely used for graph interpretation and generation, its potential has been barely explored for node and graph classification related tasks.
>
> > "have demonstrated the working mechanisms" should be removed, as the paper does not fulfil the promise made by the authors.
>
> We have revised that description to more clearly express what we meant for “working mechanisms.” In particular, we have changed it to “have demonstrated how it helps enhance the discriminative representation power …..”
>
>
> **Introduction**
>
> > Missing citations to GNN (Scarselli et al, 2009) and NN4G (Micheli, 2009)
>
> Thank you for your suggestion. But our work mainly focus on convolution-based graph neural networks (i.e., GCNs) rather than general graph neural networks.
>
> > ​​From "the absence of relational inference" to the end of the paragraph: this statement looks like pure speculation unless supported by scientific evidence or concrete examples.
>
> We have revised that description into “the absence of relational inference makes the aggregation mechanism indifferent to the multiple types of latent inter-node relations, ignoring which limits the ultimate potential of the model performance”
>
> > Lack of proper motivations for the proposed architecture
>
> Please see our “response to common concerns.”
>
> > Q: why is the GAT aggregation too "ad-hoc"? It seems pretty general and completely adaptive. And what does "ad-hoc" refer to?
>
> We intend to express that the way GAT generates the aggregation weights are not regularized by the formation of network structure, whereas in real-world networks (especially social networks) the edges are usually established as results of relations. There is a tradeoff between flexibility and relational interpretability in the aggregation weights, the methods that define relations with human knowledge enjoy relational interpretability at the cost of flexibility, and attention is the opposite. We will drop this expression in revision to avoid causing confusion.
>
> > Q: could the authors please describe how and why "extra information provided with the observed edges remedies optimization issues caused by the scarcity of label supervision"? Otherwise this seems another claim unsupported by evidence (either from the paper or from the literature).
>
> The loss of our model consists of both edge reconstruction loss and label supervision loss. Without the edge reconstruction loss, the algorithm is found to diverge when the label is scarce. We have revised this expression in the new submission to "This generative model allows the community-based relations to be inferred from both labels and edges, which not only systematically solves the relation non-observable issue, but also incorporates edges as an alternative source of information provided for the posterior inference of latent relations. Since the observed edges dominate labels in quantity, our method would not heavily depend on label supervision, hence may suffer less from the scarcity of labels."
>
>
> > Q: What are "observed (target) labels"? Are we considering a specific kind of task that considers unobserved and observed targets, like the semi-supervised setting of node classification? If that is the case, the authors should specify it, since that statement does not hold in general.
>
> We use $\yv_o$ and $\yv_u$ to distinguish the Observed and Unobserved parts of data labels.There would always be observed and unobserved labels no matter in regular supervised learning or semi-supervised learning. The observed labels are used to derive the (approximated) posterior of $\Phimat$, and such posterior will be used to predict the unobserved labels. Differentiating $\yv_o$ and $\yv_u$ is a necessity in the last paragraph of section 2.2.2.

---

> ### Author Response · Authors · 2021-11-22
> **The second part of our reply to reviewer 1iDH**
>
> $\newcommand{\Phimat}{\boldsymbol{\Phi}}$
> $\newcommand{\Amat}{\mathbf{A}}$
> $\newcommand{\Xmat}{\mathbf{X}}$
> $\newcommand{\yv}{\boldsymbol{y}}$
>
> **Methodology**
>
> > About generative and inference networks
>
>   First, the phrases we used in paper are “generative network” and “inference network”, which correspond to the structure we design for the label GENERATION process, and the structure for latent variable INFERENCE. The role / function of these networks are clearly indicated by their names. In statistical machine learning, the generative network embodies the data generation process (DGP) where data is generated from latent representations, e.g., the generator in GAN or decoder in VAE; and the inference network performs approximate posterior inference on the latent quantities, e.g., the encoder in VAE. As for the math notation of the learnable parameters in the generative and inference networks, we follow the convention in VAE [3],  a well known deep generative model. We noticed there are similar usages to what the reviewer suggested, such as in GMNN [4], but it should not be a problem as long as we make clear definitions (as we did) before reference. As a reminiscence of basic concepts in statistical machine learning, “inference” indicates the process to infer the underlying parameters/representation of the population given observed data (data ->parameters/latent variables), “generation” is the model assumption of how data is generated (latent variables -> data). Statistical inference techniques could be applied to generative models to infer the latent variables and optimize model parameters, but it is not the function of generative models. **The impression “generative models perform inference” is incorrect**.
>
> > The last statement of Sec 2.1.1 is crucial to the justification of your work, but it is again unsupported by scientific evidence.
>
> We have revised that statement as follows: “Note that the node interaction over each community uniquely corresponds to one type of inter-node relation, the generation of $\Amat$ reflects the aggregation of these heterogeneous relations, suggesting that one would need to partition the edges to separate these relations.”
>
> > Throughout the full text, there seems to be no reference whatsoever to the temperature values tried in the experiments. Please specify them.
>
> We set temperature = 1 for both node and graph classification tasks. We have specified it in the appendix of "hyperparameters" in revision.
>
> > Empirical comparison with R-GCN
>
> We find the structure of R-GCN similar to dropping the representation composer from our architecture, and we would like to keep our representation composer for the following two reasons: 1) it is a part of our model assumption to derive the label generation process, 2) empirically, we have justified the necessity of both RGCN bank and representation composer in ablation study on network structures.
>
> >There is no proof or derivations for the equations of the community encoder in Sec 2.2.1, and the analytical expression of L_KL should be described somewhere.
>
> The reparameterization of a weibull random variable is the inverse function of the CDF of the weibull distribution with shape $k$ and scale $\lambda$. We have made it clear in revision. The analytical expression of $L_{KL}$ is explicitly given in WHAI [5], we also added it to our appendix in the revised submission for easier reference.
>
> > From "and the model design that involves... to the end of the paragraph": this is another claim unsupported by scientific evidence, and it is even not very clear. It could be better to drop it or add some support to it.
>
> The paragraph for unsupervised pretraining is revised, where we add our heuristic as well as support from related works. Please refer to the revised pdf submission for details.
>
> > Since the model is end-to-end trainable, it is not completely intuitive why the supervised finetuning stage should alternate between two different supervised optimizations, one of which is included into the other.
>
> The convergence issue indeed exists, in fact we find the model diverges when directly training the inference and generative networks together with the ELBO. We will make it clear in revision. The remark “more facilitative to the classification task” is made upon the overall supervised finetune rather than the iterative training scheme, which we believe is unambiguously expressed in paper.

---

> ### Author Response · Authors · 2021-11-22
> **The third part of our reply to reviewer 1iDH**
>
> $\newcommand{\Phimat}{\boldsymbol{\Phi}}$
> $\newcommand{\Amat}{\mathbf{A}}$
> $\newcommand{\Xmat}{\mathbf{X}}$
> $\newcommand{\yv}{\boldsymbol{y}}$
>
> **Methodology**
>
> > Q: What is the meaning of "unsupervised parts of labels"?
>
> We have not used the expression “unsupervised parts of labels” in our paper, but there is a usage of “unobserved parts of labels”, which means the labels not known at the training stage.
>
> > "K positive-weighted edges" --> "K positive-weighted matrices"
>
> The positive-value matrices represent the weights on the edges, and they are the direct results of running edge partition, hence we prefer to keep the original expression "K positive-weighted edges".
>
> We appreciate your effort in pointing out the typos, we have corrected them in revision.
>
> **Empirical evaluation**
>
> We do not agree with the reviewer’s assessment “unfair, not robust, optimized wrt the test set, or all of the previous” as a major issue of **our work**. We appreciate you pointing out the two papers discussing potential drawbacks in current baselines, which we are not familiar with, but the two papers may not stand for the community’s overall view on the protocols we used, as they are widely adopted by empirical studies on node & graph classification methodologies. We have made a fair comparison with the base models we selected, because we have strictly followed the same protocol they used and selected hyper-parameters such that our architecture is comparable with our base models’. It is beyond the scope of this paper to discuss whether a protocol is a good one, and **we hope the reviews on the paper concentrate on its topic**.
>
> The comment “it is evident that … meaning the hyperparameters have been optimized on the test set” made by the reviewer is subjective. In node classifications, the optimal model has the same depths of RGCN_BANK and REPCOMP, so we keep this design for the graph classification task. Evenly partitioning a network with 5 layers, including the input layer, yields the {3}-{2} architecture. If we were optimizing Table.3 by Table.5, we would have reported better results for IMDB-B and IMDB-M.
>
>
>
> References
>
> [1] Michael Schlichtkrull, Thomas N Kipf, Peter Bloem, Rianne Van Den Berg, Ivan Titov, and Max Welling. Modeling relational data with graph convolutional networks. In European semantic web conference (ESWC), pp. 593–607. Springer, 2018.
>
> [2] Shikhar Vashishth, Soumya Sanyal, Vikram Nitin, and Partha Talukdar. Composition-based multirelational graph convolutional networks. In International Conference on Learning Representations (ICLR), 2020.
>
> [3] D. P. Kingma and M. Welling. Auto-encoding variational bayes. In Proceedings of the International Conference on Learning Representations (ICLR), 2014
>
> [4] Qu Meng, Yoshua Bengio, and Jian Tang. "Gmnn: Graph markov neural networks." In International conference on machine learning. PMLR, 2019.
>
> [5] Hao Zhang, Bo Chen, Dandan Guo, and Mingyuan Zhou. WHAI: Weibull hybrid autoencoding inference for deep topic modeling. In International Conference on Learning Representations (ICLR), 2018a.
>
> [6] Fan-Yun Sun, Meng Qu, Jordan Hoffmann, Chin-Wei Huang, and Jian Tang. vgraph: A generative model for joint community detection and node representation learning. In Advances in Neural Information Processing Systems (NeurIPS), volume 32, 2019.

---

### Official Review · Reviewer_6jg3 · 2021-11-02

**Correctness:** 3
**Technical Novelty And Significance:** 2
**Empirical Novelty And Significance:** Not applicable
**Recommendation:** 3
**Confidence:** 4

**Main Review:**

Strengths:
1) The proposed methods perform favorably in the experiments.
2) The presentation of the paper is clear and easy to follow.

Weakness:
1) The motivation of designing such partition based GNN is not clear. Why it is important to consider the latent inter-node relations? What are the implications should such relations are ignored assuming their existence?
2) It lacks a comprehensive analysis of the learned partition weights (called node community affinity matrix). While the paper provides the T-SNE plot of the learned matrix, there are not quantitative measure of the properties of learned affinity. Do these K groups of affinity scores have similar structures? Are there clear pattern of partitions when we adjust the temperature parameter?
3) The ablation study section misses a few important dimensions. a) The impact of doing pretraining step introduced in this paper. And why there are instability issues with end-to-end training. Is it sensitive to any of the parameters of the node affinity matrix? b) Does the proposed method only work with GNN?
4) The methods proposed in this paper are based mature and well-known techniques. The novelty is limited.

**Summary Of The Paper:**

This paper proposed a edge partition based graph neural network model. It samples a K sets of edges based on learnable node affinity matrixes. A composer layer is used to combine node representations from each partitions. To improve training stability, the paper trains the edge partition part first with unsupervised then finetune it on supervised learning task. Built upon mature techniques, the authors suggest this model has superior performance with experiments on both node classification and graph  classification data.

**Summary Of The Review:**

This paper uses the graph generation model and a soft-max transformation with temperature term to implement the idea of doing K graph edge partitions in side GNN computation. The presentation of paper is well organized and experiments results are generally favorable. However, the motivations and assumptions are not discussed clearly and verified carefully. The ablation study does not address some of the key design in this model. The overall novelty of this paper is limited.

---

> ### Author Response · Authors · 2021-11-22
> **We respond to the reviewer's comments and address the potential misunderstanding of our paper**
>
> Dear Reviewer 6jg3,
>
> In your review, you had flagged our paper for ethical concerns: “Flag For Ethics Review: NO., Yes, Discrimination / bias / fairness concerns.” We are very confused about this flag as you have provided no justifications. Would you please help explicitly point out your concerns on discrimination / bias / fairness ?
>
> Reply to the “weakness” comments:
>
> * Motivation:
>
>   Ignoring the latent relations may lead to two drawbacks. From the embedding propagation perspective, representations from different relations are entangled, which undermines model interpretability. From the feature transformation perspective, process relations separately enables independent transformations of node features, so the representations learned with each relation could be optimized with more degrees of freedom, which contributes to enhanced model performance. In other words, ignoring these latent relations, which can be extracted from the observed graphs, limits the ultimate potential of the model performance. Please refer to our general comment on “the motivation of method / architecture” as well as our revised introduction (in the revised pdf submission).
>
> * Assessment of partitioned graphs:
>
>   We have replied to the concern to the quality of community learning in our general comment, please refer to our response to common concerns for detail.
>
> * Ablation study on unsupervised pretrain:
>
>   We will add a comprehensive ablation study to examine the impact of the pretraining step. Here we have provided a few examples in our general comment. We could see from the table that the unsupervised pretraining step generally promotes model performances on both node and graph classification tasks. We are not quite sure we fully understand your question:  “b) Does the proposed method only work with GNN?” We appreciate it if you could elaborate it further.
>
> * We’d like to emphasize our novelty in modeling unobserved relations in a relational graph by a generative model and optimize the model in an end-to-end variational inference framework, even though no prior work has explored viewing node / graph classification problems from this perspective. Even though many modules have been proposed before, they appear in distinct contexts. They are mature techniques for their specific original purposes, but combining them into a coherent modeling framework to clearly enhance the discriminative powers for node and graph representations, which takes significant time and effort, is not a simple task at all.
>
> Given the potential misunderstanding of the paper’s main perspective (see reply to “weakness” comment 2) and contribution (see reply to “weakness” comment 4), we hope the reviewer could take another look at our paper and reassess its merits.

---

### Official Review · Reviewer_tuFU · 2021-11-02

**Correctness:** 3
**Technical Novelty And Significance:** 3
**Empirical Novelty And Significance:** 3
**Recommendation:** 5
**Confidence:** 3

**Main Review:**

Overall, this paper is well-organized. It is interesting to investigate how an edge is formed to make use of the inter-node relation to enhance GNNs for node representation learning. Experimental results on several graphs demonstrate the effectiveness of the proposed method in both node-level and graph-level tasks.

However, I have some major concerns:
- Lack of theoretical analysis. There is no theoretical analysis of the proposed method especially on the relationship between community information and embedding propagation.
- Studies of different relations. Although the aggregated representations of all relations have been visualized in Figure 2, it may provide more information to explore each relation in contributing to form edges and learning representations.
- Performance of community learning. Although in the visualization some community structures have been shown in Figure 2, it would be interesting to show quantitatively the performance of community detection.

There are some minor comments:
- Why Community ENCoder is part of model training?
- Notation r has been used as parameters on Page 3 and also as relations on Page 4.
- what does RHS mean under Eq (3)?
- There are some typos:  (1) In REPresentataion COMPoser: denotes -> denote, after Eq (3), Where -> where

**Summary Of The Paper:**

This paper investigates how an edge is formed by different latent inter-node relations and extends the community-based edge formulation mechanism to graph neural networks. A variational inference framework has been proposed to jointly learn how to partition the edges into different communities and combine relation-specific GCNs for the end classification tasks. Experiments on several real-world graph datasets demonstrate the effectiveness of the proposed method in both node-level and graph-level representation learning problems.

**Summary Of The Review:**

The paper provides an interesting and novel solution to learn better representation with the investigation of edge formulation. However, several claims are not well-explained.

---

> ### Author Response · Authors · 2021-11-22
> **We respond to the reviewer's concerns and address the potential misunderstanding of our paper**
>
> $\newcommand{\Phimat}{\boldsymbol{\Phi}}$
> $\newcommand{\Amat}{\mathbf{A}}$
> $\newcommand{\yv}{\boldsymbol{y}}$
>
> We appreciate the feedback from reviewer tuFU. There appears to be some misunderstandings on the generative model part. In the first minor comment, the reviewer asks “why community encoder is part of model training”, implying that the “correct model” would separate latent community inference and classification. We would like to emphasize that in our generative model, both the graph structure and labels are “results” of latent relations, so they are all taken into consideration in relational inference, i.e., the community encoder is expected to approximate the posterior $p(\Phimat | \Amat, \yv_o)$. Removing the community encoder from model training will make it to approximate a different posterior $p(\Phimat | \Amat)$. The design of generative model and the variational inference optimization scheme is the key difference between the proposed model and previous partition-based models (R-GCNs [1, 2] or disentangle-based models [3, 4]) or a simple combination of community detection and relational learning (the misunderstood model), therefore such misunderstanding may lead to significant under-assessment of our technical novelty. We hence request the reviewer to reconsider the review opinion on our paper.
>
> Reply to major concerns:
>
> * Theoretical analysis:
>
>   This paper is methodological focused and algorithm design driven. It would have no harm to add theoretical analysis, but would you please specify which part of model design could benefit from theoretical analysis? We are not sure we fully understand what the theories about “relationship between community information and embedding propagation” are meant to prove, would you please elaborate that?
>
> * Studies of different relations:
>
>   Our expectation for relational learning is that the community GCN bank extracts “good and diverse” relation-specific sub-representations. Our t-SNE plots qualitatively assess the diversity of learned sub-representations. The study on marginal effects of each relation is not usually provided by previous works on heterogeneous relational graphs, but we agree it may enlighten us how “good” our sub-representations are and lead to some interesting discussions. We will look into this aspect in future studies.
>
> * Community learning:
>
>   We would like to emphasize that our main focus is classification instead of community detection. The label supervision may lead to unpredictable biases from ground-truth communities. Quantitative assessment against models specifically optimized for community detection / graph clustering will be unfair to the proposed method. However, a qualitative visualization of discovered relations / communities (such as ones in EPM [5]) may help enhance the interpretability of our model. We are glad to provide similar visualizations if the reviewer finds it necessary or beneficial.
>
> Reply to other comments:
> * Repetitive usage of notation “r”: we used different fonts to distinct the community activation level indicators from relations, we have changed the notation of community activation level indicators from “r” to “$\gamma$” to avoid potential confusion.
>
> * RHS: Right Hand Side. We will pay attention to defining the acronyms before usage.
>
> * Thank you for pointing out the typos, we have fixed them in revision.
>
>
> References
> [1] Michael Schlichtkrull, Thomas N Kipf, Peter Bloem, Rianne Van Den Berg, Ivan Titov, and Max Welling. Modeling relational data with graph convolutional networks. In European semantic web conference (ESWC), pp. 593–607. Springer, 2018.
>
> [2] Shikhar Vashishth, Soumya Sanyal, Vikram Nitin, and Partha Talukdar. Composition-based multirelational graph convolutional networks. In International Conference on Learning Representations (ICLR), 2020.
>
> [3] Jianxin Ma, Peng Cui, Kun Kuang, Xin Wang, and Wenwu Zhu. Disentangled graph convolutional networks. In International Conference on Machine Learning (ICML), volume 97 of Proceedings of Machine Learning Research, pp. 4212–4221, 2019.
>
> [4] Yiding Yang, Zunlei Feng, Mingli Song, and Xinchao Wang. Factorizable graph convolutional networks. In Advances in Neural Information Processing Systems (NeurIPS), volume 33, pp. 20286–20296, 2020.
>
> [5] Mingyuan Zhou. Infinite edge partition models for overlapping community detection and link prediction. In International Conference on Artificial Intelligence and Statistics (AISTATS), volume 38 of Proceedings of Machine Learning Research, pp. 1135–1143, 2015.

---

### Official Review · Reviewer_LpaE · 2021-11-16

**Correctness:** 4
**Technical Novelty And Significance:** 4
**Empirical Novelty And Significance:** 3
**Recommendation:** 8
**Confidence:** 3

**Main Review:**

**Strengths :**

1. Novel technique. While community detection has been explored before, using this community information to predict better labels is something that I have not come across. The model training procedure with the unsupervised pretraining followed by iterative supervised learning is interesting.
2. Thorough evaluation across 11 datasets to demonstrate that the model works well in practice.

**Weaknesses :**

1. Problem needs to be better motivated. What are some examples of real world datasets with latent relations/communities? For example, what could be the latent communities in a citation network? I made an attempt to construct an example in the summary section above based on my understanding, but it would be nice to see something of the sort in the introduction.

2. The ablation studies aren’t very comprehensive - they only compare across a) different input features, b) different numbers of latent communities, and c) different numbers of GCN layers. A better ablation study, in my opinion, would be to modify key parts of the model and not just change hyperparameters. Example - what is the performance with and without the unsupervised pretrain? What if we didn’t compose $X$ with $\Phi$ to get $X*$?

More thoughts / Questions for the authors :

1. In the Edge Partitioner module, is $\tau$ always chosen such that the generated adjacency matrices are binary? I assume that must be so, because otherwise one cannot apply a GCN to a non-binary adjacency matrix. But it would be nice to have this clarified.

2. Do all the baselines use hand-crafted features as in Xu et al (2019)? I couldn’t find this mentioned anywhere.

3. Here is a thought - what if we were to take a multi-relational graph, homogenize the edges, then see what latent relations are predicted by your model and how well they track the original relations that we started with?

Minor comments :

1. It would be nice if the $||$ notation for concatenation could be explicitly defined somewhere. It becomes obvious after a little more reading, but it confused me the first time.

2. Typos - “interatively” (page 5), “represnetation” (page 7), “edge-formation contributionedge-formation contribution” (page 9).

**Summary Of The Paper:**

This paper develops Edge-Partition Modulated Graph Convolutional Networks, a GNN architecture that explicitly models latent relations between nodes in a graph. The key idea is that nodes may be organized as overlapping "communities", where each community represents a particular type of relation. For example, in a citation network of papers, where the task is to predict the category of each paper, all papers from a research group could form one latent community. If one could infer this latent community information, then this could help predict unknown labels.

To accomplish this task, the model learns a community affiliation matrix whose entries express how strongly each node is affiliated to a particular community. Once we have this matrix, the adjacency matrix is partitioned as a sum of K adjacency matrices, where K is the number of communities. The model then runs K different GCN models using the same node features but K different adjacency matrices, and the outputs of all these GCN models are concatenated. Finally, we apply a GCN model using these composite node features and the __original__ adjacency matrix, to get a final classification for each node. These GCN models together comprise the "generative network".

The different weights are learned using an intricate training procedure. First, the model learns a good initial community affiliation matrix, by maximizing the posterior probability of the community matrix given the adjacency matrix. This stage is entirely unsupervised, because it uses only the observed edges and not the edge labels.

In the next stage, the model iteratively runs inference (where the community affiliation matrix is learned by optimizing the ELBO using label supervision) and learning (where the community matrix is held fixed but the generative GCN models are trained). Inference and learning are repeated till convergence.

Finally, a community matrix is sampled from the learned posterior and this is used along with the generative network to predict unknown labels.

The authors evaluate their model on a variety of node classification and graph classification benchmarks. They also visualize some embeddings for better interpretability, and perform some ablation studies.


**Summary Of The Review:**

Overall this paper makes an interesting contribution to the field. The technique/key idea are, in my opinion, novel, and the evaluation is thorough. My concerns are limited to the problem motivation as presented in the paper, and the extent of the ablation studies.

It is possible that my understanding of the variational inference and ELBO optimization part is insufficient. I have not checked these parts for correctness, and I am not familiar with related work in the area. I have therefore indicated that my confidence in my assessment is slightly lower.

---

> ### Author Response · Authors · 2021-11-22
> **Reply to reviewer LpaE**
>
> $\newcommand{\Phimat}{\boldsymbol{\Phi}}$
> $\newcommand{\Amat}{\mathbf{A}}$
> $\newcommand{\Xmat}{\mathbf{X}}$
> $\newcommand{\given}{\,|\,}$
> $\newcommand{\yv}{\boldsymbol{\y}}$
>
> We appreciate the valuable suggestions and the overall positive feedback given by the reviewer LpaE. We are glad that you recognized the novelty of our method in dealing with multiple latent communities.
>
> Reply to the “weakness” comments:
>
> * Motivation: we consider adding the following example to our paper. In a social network where people are linked by online friendship, one person may be simultaneously affiliated with multiple social groups (e.g., graduating from the same school A, or working for the same company B, etc.). The co-membership of a social group between a person and his/her friends naturally forms one type of similarity-based relations (e.g., schoolmates of A, colleagues in B). We also highlighted the motivation of the partition-based method in our general comment. The example is added in our rebuttal revision.
>
> * Ablation studies: We have performed the ablation studies on input features in section 4.5 under “various input features”, which includes the comparison between $\Xmat^*=\Xmat$ and $\Xmat^* = \Xmat \Vert \Phimat$. Also in table 4, this comparison is made in the second and the fourth rows. We will add the ablation study for unsupervised pretrain in our revision, we added a few examples to show the impact of unsupervised pretraining in our general comment.
>
> Clarification of some questions:
>
> * We would like to point out that the binary adjacency matrix is not mandated by the embedding propagation function of GCN or its practices to normalize the adjacency matrix, although a near binary partition would definitely lead to better interpretability (because all edges are tagged with only one relation instead of a linear combination of all relations). In practice, we select $\tau=1$ for node classification experiments by cross-validation, and use the same value for graph classification experiments. We have made the value of $\tau$ clear in the revised version.
>
> * Yes, all of the base models we cited in graph classification follow the same protocol as Xu et al (2019).
>
> * This is a very good suggestion, comparing our inferred relation against some groundtruth relation would definitely lead to interesting results. However, we would like to point out that the latent community inferred by EPM-GCN are not only used to explain the generation of edges, but also optimized by the label supervision, which may lead to unpredictable bias from the ground truth community partition. As for the interpretability of our discovered relations, we are glad to provide visualizations similar to EPM [1] (please refer to our general comment) if the reviewer finds them necessary or beneficial.
>
> Finally, we have followed your suggestion on clarifying the math notation for concatenation and carefully proofread the manuscript to correct the typos. We appreciate your effort to point them out.
>
> [1] Mingyuan Zhou. Infinite edge partition models for overlapping community detection and link prediction. In International Conference on Artificial Intelligence and Statistics (AISTATS), volume 38 of Proceedings of Machine Learning Research, pp. 1135–1143, 2015.

---

### Author Response · Authors · 2021-11-22
**Response to common concerns**

We appreciate all reviewers for the comments and suggestions. Here we would like to clarify some concerns shared by more than one reviewer.

* The motivation of method / architecture (LpaE, 6jg3, 1iDH):

  We find partitioning the graph into multiple graphs by relation and learning each relation with a separate GCN a standard practice to deal with graphs with heterogeneous relations in previous works [1, 2], which are related to and help provide justifications for our unique latent-community based design of the label generation process. Note in [1,2], the edges have labels and hence a separate GCN can be naturally created for each edge label. In our paper, we “soft” partition each edge into multiple latent communities, which can then be used to create multiple community-dependent GCNs without using edge labels.

  **We have revised our paper** to make sure our motivation highlighted, please refer to our revised submission for details. **We use blue text color to highlight the revision we have made**.


* The ablation study for unsupervised pretraining (LpaE,6jg3):

  We will add a comprehensive ablation study to examine the impact of the pretraining step. Here we’d like to first provide a few examples (the results for cora and citeseer datasets are node classification accuracies, and the results for MUTAG and PTC datasets are graph classification accuracies)

  | dataset | cora | citeseer | MUTAG | PTC |
  | :-: | :-: | :-: | :-: | :-: |
  | acc (with pretrain) | 84.0 $\pm$ 0.1 | 72.4 $\pm$ 0.1 | 93.6 $\pm$ 3.4 | 75.6 $\pm$ 0.9 |
  | acc (without pretrain) | 81.7 $\pm$ 0.1 | 70.5 $\pm$ 0.1 | 86.6 $\pm$ 6.6 | 68.6 $\pm$ 2.0 |

  From the above examples we could see that the performance generally drops if the unsupervised pretraining stage is not included in the training scheme.

* The quality of community learning by the proposed method (LpaE, tuFU, 6jg3)

  We would like to emphasize that the model is not developed for community discovery per se, as our expectation for these graph factors is to help the community GCN bank to produce “good and diverse” community-specific sub-representations to add more discriminative power to the composed node representation. The t-SNE on sub-representations is a more direct justification of its effectiveness for enhancing discriminative powers for node/graph classifications. We also note the edge partition model of [4], which is incorporated into our classification models, has already provided rich visualization of the discovered latent communities. We’d be glad to provide similar visualizations if the reviewers consider it as necessary or beneficial.

References

[1] Michael Schlichtkrull, Thomas N Kipf, Peter Bloem, Rianne Van Den Berg, Ivan Titov, and Max Welling. Modeling relational data with graph convolutional networks. In European semantic web conference (ESWC), pp. 593–607. Springer, 2018.

[2] Shikhar Vashishth, Soumya Sanyal, Vikram Nitin, and Partha Talukdar. Composition-based multirelational graph convolutional networks. In International Conference on Learning Representations (ICLR), 2020.

[3] M. Nickel, K. Murphy, V. Tresp, and E. Gabrilovich. A review of relational machine learning for knowledge graphs. Proceedings of the IEEE, 104(1):11–33, Jan 2016. ISSN 0018-9219. doi: 10.1109/JPROC.2015.2483592.

[4] Mingyuan Zhou. Infinite edge partition models for overlapping community detection and link prediction. In International Conference on Artificial Intelligence and Statistics (AISTATS), volume 38 of Proceedings of Machine Learning Research, pp. 1135–1143, 2015.

---

### Public Comment · ~XU_HAN6 · 2022-01-29
**Question for pre-training**

Hi,
I have a question about the pre-training stage. Is the model only pre-trained on the target dataset? Is there any other extra data used? Since the pre-training process is always on a large dataset, and then the model will be fine-tuned to the downstream task. I noticed actually the dataset for graph classification is quite small, so I am not sure if it is enough for unsupervised learning.

Thanks!

---

> ### Public Comment · ~Yilin_He1 · 2022-01-29
> **Data for pretraining**
>
> Thank you for your interest in our work. The data for pretraining is the same with the training set used in supervised finetune. Only the community encoder is involved in the pretraining stage, which aims to reconstruct the graph(s) in the training set. The observations that could be used for this particular optimization problem would be the connection statuses (connected / disconnected) of all node pairs in all graphs in the training set, whose total number would be sufficient for training the 2-layer community encoder. From our experiments, we notice that the values of the loss function for pretraining ($L_{rec}$ + $L_{KL}$) always converge at the end of pretraining stage.

---

> > ### Public Comment · ~XU_HAN6 · 2022-01-30
> > **Question for pre-training**
> >
> > Thanks for your reply! I think the reported result is impressive. How is the time for the pre-training process compared with supervised learning? Will it cause some trouble if   the model is applied to the large dataset such as ogb?

---

> > > ### Public Comment · ~Yilin_He1 · 2022-01-31
> > > **Time consumption of pretraining v.s. supervised training**
> > >
> > > Thanks for the follow-up. The time consumption for pretraining is higher than supervised training due to more training iterations. For a single iteration in pretraining and supervised training, the duration is comparable. We think it is because the most time-consuming loss term, namely $L_{rec}$, exists both in the pretraining loss and the supervised training loss. It is the key term that incorporates information from graph structure into supervised learning.
> > >
> > > Speaking of applying the algorithm to large datasets, we have not tried obg yet. The largest graph in our experiments is PubMed. For this particular dataset, due to the limitation of GPU memory, to run forward as well as backward computations regarding $L_{rec}$, we have to break the adjacency matrix into blocks, compute $L_{rec}$ on each block then aggregate. I assume the same issue would happen to obg if the graph size is larger than PubMed. A workaround could be sub-sampling the graph before calculating $L_{rec}$, we consider include that in our future work.

---

> > > > ### Public Comment · ~XU_HAN6 · 2022-01-31
> > > > **Question for pre-training**
> > > >
> > > > Thanks for your patience! I like this work and good luck with the future submission.

---

### Decision · Program_Chairs · 2022-01-20

**Decision:**

Reject

**Comment:**

The weaknesses of the paper can briefly be summarised as follows: i) the suggested motivation is not so clear, and in addition the experimental results (by themselves questionable in the way they are obtained) do not support the main claim of the paper that "...edges are generated by aggregating the node interactions over multiple overlapping node communities, each of which represents a particular type of relation that contributes to the edges via a logical OR mechanism." In fact, the observed separation among components is not proven to be of the predicted nature. ii) empirical results are obtained using a deprecated experimental protocol. For the field to make real progress, experimental assessments should follow statistically sound protocols. Already published papers that were not following a sound protocol should not be taken as reference for future empirical assessments.
The last point alone is a strong motivation for rejecting the paper.